# Information Theoretic Counterfactual Learning from Missing-Not-At-Random Feedback

**Zifeng Wang**[1],**Xi Chen**[2],**Rui Wen**[2],**Shao-Lun Huang**[1],**Ercan E. Kuruoglu**[1,3],**Yefeng Zheng**[2]

[1]Tsinghua-Berkeley Shenzhen Institute, Tsinghua University      [2]Jarvis Lab, Tencent
[3]Institute of Science and Technologies of Information, CNR, Pisa, Italy
wangzf18@mails.tsinghua.edu.cn    {jasonxchen,ruiwen,yefengzheng}@tencent.com
shaolun.huang@sz.tsinghua.edu.cn    ercan.kuruoglu@isti.cnr.it

## Abstract

Counterfactual learning for dealing with missing-not-at-random data (MNAR) is an intriguing topic in the recommendation literature since MNAR data are ubiquitous in modern recommender systems. Missing-at-random (MAR) data, namely randomized controlled trials (RCTs), are usually required by most previous counterfactual learning methods for debiasing learning. However, the execution of RCTs is extraordinarily expensive in practice. To circumvent the use of RCTs, we build an information-theoretic counterfactual variational information bottleneck (CVIB), as an alternative for debiasing learning without RCTs. By separating the task-aware mutual information term in the original information bottleneck Lagrangian into factual and counterfactual parts, we derive a contrastive information loss and an additional output confidence penalty, which facilitates balanced learning between the factual and counterfactual domains. Empirical evaluation on real-world datasets shows that our CVIB significantly enhances both shallow and deep models, which sheds light on counterfactual learning in recommendation that goes beyond RCTs.

## 1 Introduction

A surge of research shows that the real-world logging policy often collects missing-not-at-random (MNAR) data (or selective labels) [25]. For example, users tend to reveal ratings for items they like, thus the observed users' feedback, usually described by click-through-rate (CTR), can be substantially higher than those not observed yet. Consider a mini system with two users and three items in Fig. 1, when ignoring the unobserved events, the estimated average CTR from the observed outcomes is $2/3 \approx 0.67$. However, if the rest unobserved outcomes were 0, the true CTR would be $1/3 \approx 0.33$. This gap between the *factual* and *counterfactual* ratings in MNAR situation further exaggerates due to path-dependence that the learned policy tends to overestimate on the observed events outcomes [32].

Table 1: Missing ratings in a recommender system, where ✓, ✗ and ? mean positive, negative and unknown outcomes, respectively.

|        | Item 1 | Item 2 | Item 3 |
|--------|--------|--------|--------|
| User A | ✓      | ?      | ?      |
| User B | ✗      | ?      | ✓      |

Vast majority of existing works in the recommendation literature neglect the MNAR effect, as they mainly focus on designing novel architectures and training techniques for improving model performance on the observed events [8, 12, 24], where the objective function is designed in principle

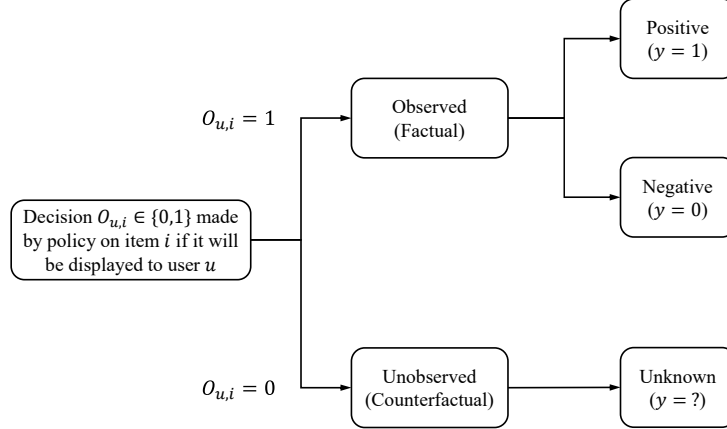

Figure 1: $O$ decides the appearance of the event, and depends on the previous recommendation policy.

of empirical risk minimization (ERM) as

$$\mathcal{L}_{ERM} = \frac{1}{N_l} \sum_{(u,i):O_{u,i}=1} \ell_{u,i}(\hat{Y}, Y). \tag{1}$$

$x = (u, i)$ is an event composed of user $u \in \mathcal{U}$ and item $i \in \mathcal{I}$; $O_{u,i} \in \{0, 1\}$ indicates whether the outcome of an event $x = (u, i)$ is observed (i.e., whether item $i$ is presented to user $u$); $\ell_{u,i}(\hat{Y}, Y)$ is the loss function taking true outcome $Y$ and predicted outcome $\hat{Y}$ as its inputs; and $N_l$ and $N_{ul}$ are the numbers of observed and unobserved events, respectively. However, $\mathcal{L}_{ERM}$ is not an unbiased estimate of the true risk $\mathcal{L}$ [32]:

$$\mathbb{E}_O[\mathcal{L}_{ERM}] \neq \mathcal{L} := \frac{1}{N_l + N_{ul}} \sum_{u,i} \ell_{u,i}(\hat{Y}, Y), \tag{2}$$

which indicates that the naive ERM-based method cannot guarantee model's generalization ability on the counterfactuals.

As the distribution of $O$ depends on the deployed recommendation policy at past, as shown in Fig. 1, we can regard it a representative of the *policy bias*, which influences the distribution of factual events and then the learned policy. In order to alleviate the policy bias, there are a series of works emphasizing on employing randomized controlled trials (RCTs) [6] to collect the so-called *unbiased* dataset. By adopting a uniform policy that randomly displays items to users, the logged feedback can be regarded as missing at random (MAR), which is consistent with the underlying joint distribution $p(x, y)$. Therefore, one can either evaluate the model's true generalization ability on MAR data [23], or utilize MAR data to debias learning via importance sampling [30]. Besides, Rosenfeld et al. [29] and Bonner and Vasile [5] proposed to employ domain adaptation from MNAR data to MAR data, in order to balance the predictive capability of the learned model over the factuals and counterfactuals. However, RCTs are extraordinarily expensive to be executed in a real-world recommender system. It is tricky because no solid theoretical definition of how large RCTs would be representative enough. It is questionable if small RCTs, compared with the enormous quantity of possible events, can lead to a proper estimate of the users' true preference distribution.

Considering the above challenges, we try to mitigate MNAR sampling bias via an RCT-free paradigm. Different from the domain adaption based methods, we propose an information theoretic approach for learning a balanced model based on Information Bottleneck (IB) [35, 36], which is promising for learning a representation that is informative to labels and robust against input noise [2]. In particular, we extend the original IB to counterfactual learning with a variational approximation, termed **C**ounterfactual **V**ariational **I**nformation **B**ottleneck (CVIB). The principle of CVIB is to encourage the learned representation to be *equally informative* to both the factual and counterfactual outcomes, by specifying a contrastive information in order to improve the learned embeddings generalization ability on the counterfactual domain. Besides, the minimality term in CVIB contributes to pushing the embeddings *independent* of the policy bias. In summary, our contributions are as follows:

- We establish a novel CVIB framework adapted from IB, which suggests new avenues in counterfactual learning from MNAR data in an RCT-free paradigm.
- A novel solution is proposed to handle the optimization of CVIB on MNAR data, which specifies a contrastive information term associated with a minimality term.
- We empirically investigate our method's advantages on real-world datasets for correcting MNAR bias without the need of acquiring the RCTs.[1]

## 2  Counterfactual Variational Information Bottleneck

In this section, we start from the conception about the information bottleneck. Then, we propose our novel CVIB objective function by separating events into factual and counterfactual domains. After that, we provide insights in the interpretation of the minimality term in our CVIB.

### 2.1  Problem Setup

Embedding is a conceptually classical approach for modeling the user and item in rating prediction. For example, in collaborative filtering [22], it represents an event $x = (u, i)$ by concatenation as $z = (e_u, e_i)$, then generates outcome prediction by $\hat{y} = e_u^\top e_i$. In deep learning models [13], there are multiple hidden layers $h_1, \ldots, h_L$ sequentially processing the embedding $z$, such as $h_1 = W^{(1)} z$ and $h_l = W^{(l)} h_{l-1}$ for $l = 2, \ldots, L$. In this work, we view the feedforward neural network layers as a Markov chain of successive representations, indicated with

$$y \to x \to z \to h_1 \to \cdots \to h_L \to \hat{y}, \tag{3}$$

where $y$ is the true feedback returned by users (may be unknown to our learning algorithm). According to Tishby et al. [36], a standard information bottleneck has the following form

$$\mathcal{L}_{IB} := \beta I(z; x) - I(z; y), \tag{4}$$

where $I(\cdot; \cdot)$ denotes mutual information of two random variables, and $\beta$ is a Lagrangian multiplier. Minimizing $I(z; x)$ encourages the representation $z$ to be compressed, i.e., being *minimal* to $x$. And, maximizing the second term $I(z; y)$ encourages $z$ to be sufficient to task $y$, such that $z$ can be predictive of $y$. Essentially, $\mathcal{L}_{IB}$ in Eq. (4) can be regarded as a supervised loss term plus an additional information regularization term on the representation [1]. Moreover, there is another random variable $O$ that affects the appearance of events $x$, but is not informative to the true outcomes $y$, i.e., $O \perp\!\!\!\perp y$. In this scenario, we would like $z$ to be independent of $O$, thus being free of policy bias.

The main challenge in adopting IB for optimization is that the mutual information in $\mathcal{L}_{IB}$ is cumbersome for calculation. Although previous works try to derive computable proxy for specific tasks [3], they are not suitable for MNAR data. Since we only have *partial feedback* about $y$, i.e., the majority of events are counterfactuals, we have no access to their true outcomes. Next, we will turn to derive a new objective function addressing this challenge.

### 2.2  Building Contrastive Information Regularizer

For conciseness, we focus on a simple model with only the embedding layer $z$, namely $x \to z \to \hat{y}$, and extend it to multi-layer scenario in Section 2.3. Specifically, the second term in Eq. (4) is mutual information between embedding $z$ and target task $y$. We separate the embeddings into two parts: $z^+$ and $z^-$, which represent factual and counterfactual embeddings, respectively, i.e., $z^+ \sim p(z|x^+)$ and $z^- \sim p(z|x^-)$.

As shown in Fig. 2, we factorize the original mutual information term by $I(z; y) = I(z^+, z^-; y)$. We postulate that $z^+$ and $z^-$ are independent, therefore according to the chain rule of mutual information:

$$I(z^+, z^-; y) = I(z^+; y|z^-) + I(z^-; y) = I(z^+; y) + I(z^-; y). \tag{5}$$

However, as the outcomes $y$ of the counterfactuals are unknown, we have to identify another refined solution. Specifically, we cast Eq. (5) to

$$I(z^+; y) + I(z^-; y) = \underbrace{I(z^-; y) - I(z^+; y)}_{\text{contrastive}} + 2I(z^+; y) \tag{6}$$

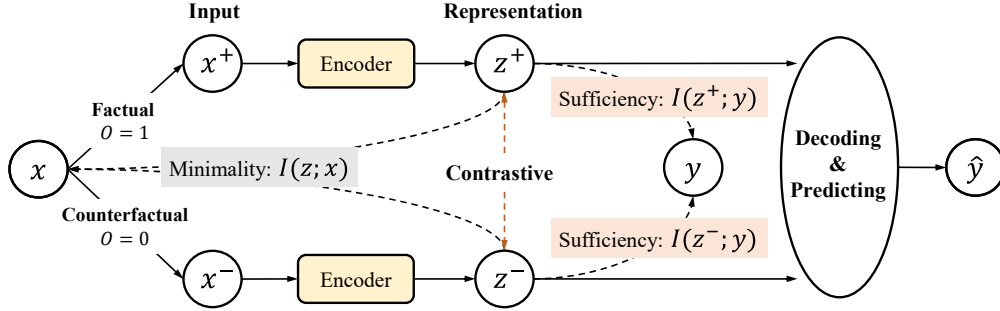

Figure 2: The events and embeddings are separated by $O$, and we introduce additional constrastive scheme between $z^+$ and $z^-$.

from which we derive a contrastive term between $z^+$ and $z^-$. This characterization is helpful for us to introduce a hyperparameter $\alpha$ to control the degree of this contrastive penalty. We then rewrite the original IB loss to

$$\mathcal{L}_{CVIB} := \beta I(z;x) + \alpha[I(z^+;y) - I(z^-;y)] - I(z^+;y) \qquad (7)$$

as our new objective function, where we propose an information theoretic regularization on $z^+$ and $z^-$. Intuitively, minimizing this term corresponds to encouraging $z^+$ and $z^-$ to be equally informative to the targeted task variable $y$, thus resulting in a more balanced model. More importantly, it does not need access to the counterfactual outcomes, which will be specified in Section 3.2.

## 2.3 Minimal Embedding Insensitive to Policy Bias

Aside from the task-aware mutual information $I(z;y)$ in Eq. (7), $I(z;x)$ corresponds to the minimality of the learned embedding. Recall that we assume that the event $x$ follows the generative process $p(x, O) = p(O)p(x|O)$, where $O$ influences the appearance of $x$. Because $O$ is independent to task $y$, we hope the predicted outcome $\hat{y}$ is not influenced by $O$, or the learned embedding $z$ should contain low information about $O$. In this viewpoint, following the practice by Achille and Soatto [1], we identify that the minimality term is actually beneficial for embedding's insensitivity against the nuisance $O$.

**Proposition 1** (Minimal Representation Insensitive to Policy Bias). *With the Markov chain assumption defined by Eq. (3), for any hidden embedding $h_k$, we can derive the upper bound of the $I(h_k; O)$*

$$I(h_k; O) \leq I(h_k; x) - I(x; y) \leq I(z; x) - I(x; y) \leq I(z; x), \qquad (8)$$

*where the last term $I(x; y)$ is a constant with respect to the training process.*

Please refer to Appendix A for a proof. Above proposition implies that an embedding $z$ is insensitive to the policy bias $O$, by simply reducing $I(z;x)$. Meanwhile, by maximizing $I(z;y)$ in IB Lagrangian, embedding $z$ will be forced to retain minimum information from $x$ that is pertinent to the task $y$. In deep models, according to the Data Processing Inequality (DPI) [9], minimizing $I(z;x)$ also works for controlling the policy bias of the successive layers.

## 3 Tractable Optimization Framework

The proposed $\mathcal{L}_{CVIB}$ is still intractable for optimization. In this section, we attempt to find a tractable solution for three terms in $\mathcal{L}_{CVIB}$, respectively. And, we present our algorithm of learning debiased embeddings by $\mathcal{L}_{CVIB}$ at last.

### 3.1 Minimality Term

The minimality term $I(z;x)$ in Eq. (7) can be measured with a Kullback-Leibler (KL) divergence as

$$I(z;x) = \mathbb{E}_x[\text{KL}(p(z|x) \| p(z))] = \mathbb{E}_x\left[\int p(z|x)\log p(z|x)dz - \int p(z)\log p(z)dz\right]. \qquad (9)$$

To avoid operating on the marginal $p(z) = \int p(z|x)p(x)dx$, we use a variational approximation of $q(z)$ as the marginal $p(z)$, which renders

$$-\int p(z)\log p(z)dz \le -\int p(z)\log q(z)dz \Rightarrow \text{KL}(p(z|x) \parallel p(z)) \le \text{KL}(p(z|x) \parallel q(z)). \quad (10)$$

Suppose the posterior $p(z|x) = \mathcal{N}(e(x); \text{diag}(\sigma))$ is a Gaussian distribution, where $e(x)$ is the encoded embedding of input event $x$ and $\text{diag}(\sigma)$ indicates a diagonal matrix with elements $\sigma = \{\sigma_d\}_{d=1}^D$. In other words, we assume the embedding is generated by

$$z = e(x) + \varepsilon \odot \sigma \quad \text{where } \varepsilon \sim \mathcal{N}(0, I). \quad (11)$$

If we fix $\sigma_d = 0$, $\forall d$, then $z$ would default to a deterministic embedding. Moreover, by considering a standard Gaussian variational marginal $q(z) = \mathcal{N}(0, I)$, the KL term reduces to

$$\text{KL}(p(z|x) \parallel q(z)) = \|e(x)\|_2^2 + \sum_d \left(\sigma_d - \frac{1}{2}\log\sigma_d\right) - D, \quad (12)$$

which means for a deterministic embedding, minimizing term $I(z;x)$ is equivalent to directly applying $\ell_2$-norm regularization on the embedding vector.

## 3.2 Contrastive Mutual Information Term

The mutual information $I(z;y) = H_p(y) - H_p(y|z)$, where $H_p(\cdot)$ demotes the entropy of $p(\cdot)$. The first entropy term is constant, which means maximizing $I(z;y)$ is equivalent to minimizing the second term $H_p(y|z)$. We then further derive the lower bound of $-H_p(y|z)$ as[2]

$$
\begin{aligned}
I(z;y) &= \int\int p(y,z)\log p(y|z)dzdy + \texttt{const} \\
&\ge \int\int p(y,z)\log q(y|z)dzdy + \texttt{const} = -H_{p,q}(y|z) + \texttt{const}.
\end{aligned}
\quad (13)
$$

The term $q(y|z)$ is an estimate of $p(y|z)$ with a classifier parameterized by $\theta$, e.g., weight matrices in deep networks or embedding parameters. We can use the cross entropy as a proxy for the mutual information in the IB objective [1, 15], because $\max I(z;y) \Leftrightarrow \min H_{p,q}(y|z)$ as shown above. Therefore, replacing the contrastive term $I(z^+;y) - I(z^-;y)$ in Eq. (7) with $H_{p,q}(y|z^-) - H_{p,q}(y|z^+)$ obtains

$$H_{p,q}(y|z^-) - H_{p,q}(y|z^+) = \int\int p(y, z^+, z^-)[\log q(y|z^+) - \log q(y|z^-)]dydz^+dz^-. \quad (14)$$

Since we assume $z^+$ and $z^-$ are independent, the generative process can be written as $p(y, z^+, z^-) = p(y|z^+, z^-)p(z^+)p(z^-)$. In order to make the term tractable, we here approximate $p(y|z^+, z^-)$ by $q(y|z^+)$,[3] then cast Eq.(14) to

$$\int p(y|z^+, z^-)[\log q(y|z^+) - \log q(y|z^-)]dy \Rightarrow \int q(y|z^+)[\log q(y|z^+) - \log q(y|z^-)]dy. \quad (15)$$

This term further goes to our final results as

$$\int q(y|z^+)\log q(y|z^+)dy - \int q(y|z^+)\log q(y|z^-)dy \Rightarrow H_q(y|z^+, y|z^-) - H_q(y|z^+), \quad (16)$$

where the first term of the right hand side is the cross entropy between $q(y|z^+)$ and $q(y|z^-)$. We identify that the second term is in line with the maximum entropy principle [4, 17] and the confidence penalty proposed by Pereyra et al. [27] that is imposed on the output distribution of deep neural networks. While out of our derivation, the confidence penalty is only imposed on the factual output $q(y|z^+)$. We hence propose balanced learning by restricting the distance between factual and counterfactual posterior, which provably strengthens model's generalization over the underlying users' true preference distribution.

**Algorithm 1** Counterfactual Learning with CVIB in MNAR Data for Recommendation.

---

**Require:** Training factual set $\Omega^+$, counterfactual set $\Omega^-$; Hyperparameters $\alpha$, $\beta$, $\gamma$;
   Initialize model's parameters $\theta$;
  **repeat**
     Sample a batch of paired factuals $X^+$, $Y^+$ from $\Omega^+$, and counterfactuals $X^-$ from $\Omega^-$;
     Compute the sufficiency term ① in Eq. (18);
     Compute the balancing term ② in Eq. (18);
     Compute the confidence penalty term ③ in Eq. (18);
     Compute the minimality term ④ in Eq. (18);
     Compute the batch objective loss as $\hat{\mathcal{L}}_{CVIB} = ① + \alpha② - \gamma③ + \beta④$;
     Update the model parameters $\theta$ via stochastic gradient descent based on $\hat{\mathcal{L}}_{CVIB}$;
  **until** training loss stops to decrease.

---

### 3.3 Task-aware Mutual Information Term

We next omit the superscript of $z^+$ in $I(z^+; y)$ for simplicity in the following. Similar to the operation of task-aware mutual information in Eq. (13), we use an approximation $q(y|z)$ to substitute $p(y|z)$

$$I(z; y) \geq \mathbb{E}_{z,x}\left[\int p(y|x) \log q(y|z) dy\right] \Rightarrow -I(z; y) \leq H_{p,q}(y|z), \tag{17}$$

which indicates that this term is a proxy of the cross entropy loss between the true outcome $p(y|x)$ and the prediction $q(y|z)$.

### 3.4 Algorithm

Taking all the above derivations together, we conclude to the final objective function $\hat{\mathcal{L}}_{CVIB}$, which encompasses four terms:

$$\hat{\mathcal{L}}_{CVIB} = \underbrace{H_{p,q}(y|z^+)}_{①\ \text{Sufficiency}} + \underbrace{\alpha H_{q,q}(y|z^+, y|z^-)}_{②\ \text{Balancing}} - \underbrace{\gamma H_q(y|z^+)}_{③\ \text{Penalty}} + \underbrace{\beta(\|e(x^+)\|_2^2 + \|e(x^-)\|_2^2)}_{④\ \text{Minimality}} \tag{18}$$

where term ① denotes the cross entropy between $p(y|z^+)$ and $q(y|z^+)$; term ② is cross entropy between $q(y|z^+)$ and $q(y|z^-)$; and term ③ is entropy of $q(y|z^+)$. For computing this objective, we need to draw $x^+, y^+$ from the factual set $\Omega^+$ and $x^-$ from the counterfactual set $\Omega^-$ in one iteration. Specifically, term ① is supervised loss on the factual outcomes; terms ②, ③ are contrastive regularization for balancing between factual and counterfactual domains; and term ④ is minimality loss to improve the model's robustness against policy bias. The whole optimization process over the $\hat{\mathcal{L}}_{CVIB}$ is hence summarized in Algorithm 1, where we should specify the values of $\alpha$, $\beta$ and $\gamma$ to balance these terms in optimization.

## 4 Empirical Evaluation

Aiming to validate CVIB's effectiveness, we perform experiments on real-world datasets in this section. We elaborate on the experimental setups, and report the comparison results between CVIB and other baselines, which substantiate its all-round superiority in counterfactual learning.

### 4.1 Datasets

In order to evaluate the learned model's generalization ability on the underlying groundtruth distribution $p(x, y)$, rather than the only on the logged feedback, i.e., the factual domain, we usually need an additional MAR test set, where the users are served with uniformly displayed items, namely RCTs. As far as we know, there are only two open datasets that satisfy this requirement:

**Yahoo! R3 Dataset** [25]. This is a user-song rating dataset, where there are over 300K ratings self-selected by 15,400 users in its training set, hence it is MNAR data. Besides, they collect an additional MAR test set by asking 5,400 users to rate 10 randomly displayed songs.

Table 2: MSE and AUC on the MAR test set of COAT [30] and YAHOO [25], where the best ones are in bold.

|  | COAT | | YAHOO | |
|---|---|---|---|---|
|  | MSE | AUC | MSE | AUC |
| MF | 0.2451 | 0.7020 | 0.2493 | 0.6767 |
| +IPS [30] | 0.2299 | 0.7156 | 0.2260 | 0.6793 |
| +SNIPS [33] | 0.2374 | 0.6960 | 0.1945 | 0.6810 |
| +DR [18] | 0.2357 | 0.7058 | 0.2108 | 0.6883 |
| +DRJL [37] | 0.2423 | 0.6915 | 0.2745 | 0.6892 |
| +CVIB (ours) | **0.2189** | **0.7218** | **0.1671** | **0.7198** |
| NCF | 0.2030 | 0.7688 | 0.3313 | 0.6772 |
| +IPS [30] | 0.2008 | 0.7708 | 0.1777 | 0.6708 |
| +SNIPS [33] | **0.1922** | 0.7695 | 0.1699 | 0.6880 |
| +DR [18] | 0.2161 | 0.7514 | **0.1698** | 0.6886 |
| +DRJL [37] | 0.2097 | 0.7579 | 0.2789 | 0.6820 |
| +CVIB (ours) | 0.2017 | **0.7713** | 0.2820 | **0.6989** |

**Coat Shopping Dataset** [30]. This dataset consists of 290 users and 300 items. Each user rates 24 items by themselves, and is asked to rate 16 uniformly displayed items as the MAR test set.

We simplify the underlying rating prediction problem to binary classification in our experiments, by making rating which is 3 or higher be positive feedback and those lower than 3 as negative.

## 4.2 Baselines

Our CVIB is model-agnostic, thus applicable for most models in recommendation that take embeddings to encode the events, including the shallow and deep models. In our experiments, we pick matrix factorization (MF) [21] as shallow backbone and neural collaborative filtering (NCF) [13] as the deep backbone. Both of these methods project users and items into a shared space and represent them with unique vectors, namely embeddings.

The most popular technique to debias MNAR data by involving RCTs is the inverse propensity score (IPS) method [30]. Its variants, the self-normalized IPS (SNIPS) [33], doubly robust (DR) [18] and joint learning doubly robust (DRJL) [37] are also widely used. In our experiments, we take $5\%$ of test data to learn the propensity scores via a naive Bayes estimator [30]

$$p(O_{u,i} = 1|Y_{u,i} = y) = \frac{p(y|O = 1)p(O = 1)}{p(y)}, \qquad (19)$$

for IPS, SNIPS, DR, and DRJL. Applying these methods to both shallow and deep backbones, we involve muliple baselines in comparison with our MF-CVIB and NCF-CVIB. Note that only CVIB is RCT-free among all methods.

## 4.3 Experimental Protocol

We implement all the methods on PyTorch [26]. For both the MF and NCF, we fix the embedding size of both users and items to be 4 because in our experiments, we find when embedding size gets larger, the performance of all methods on the MAR test set decays, which may be caused by overfitting. We randomly draw $30\%$ data from the training set for validation, on which we apply a grid search for hyperparameters to pick the best configuration. Adam [20] is utilized as the optimizer for fast convergence during training, with its learning rate in $\{0.1, 0.05, 0.01, 0.005, 0.001\}$, weight decay in $\{10^{-3}, 10^{-4}, 10^{-5}\}$, and batch size in $\{128, 256, 512, 1024, 2048\}$. For NCF, we set an additional hidden layer with width 8. Specifically for CVIB, we set the hyperparameters $\alpha \in \{2, 1, 0.5, 0.1\}$, and $\gamma \in \{1, 0.1, 10^{-2}, 10^{-3}\}$. Since we already set weight decay for Adam, we do not apply the $\ell_2$-norm term on the embeddings. After finding out the best configuration on the validation set, we evaluate the trained models on the MAR test set.

Table 3: Average nDCG with 10 runs on the MAR test set of COAT and YAHOO where the best ones are in bold.

| COAT | MF | IPS | SNIPS | DR | DRJL | CVIB |
|---|---|---|---|---|---|---|
| nDCG@5 | 0.589 | 0.633 | 0.603 | 0.622 | 0.608 | **0.663** |
| nDCG@10 | 0.667 | 0.689 | 0.676 | 0.693 | 0.679 | **0.721** |
| YAHOO | MF | IPS | SNIPS | DR | DRJL | CVIB |
| nDCG@5 | 0.633 | 0.636 | 0.635 | 0.659 | 0.652 | **0.734** |
| nDCG@10 | 0.762 | 0.760 | 0.762 | 0.774 | 0.770 | **0.820** |

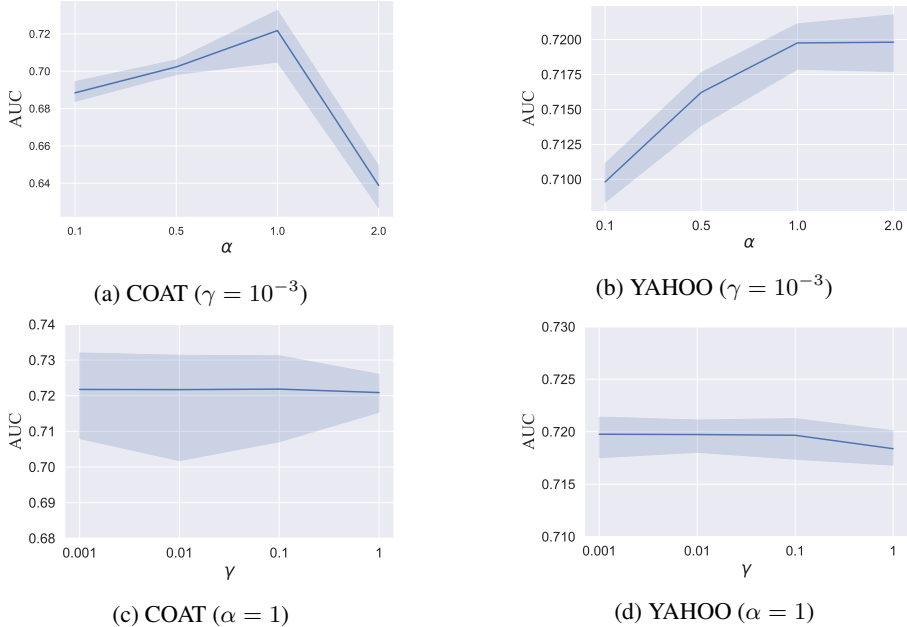

(a) COAT ($\gamma = 10^{-3}$)  (b) YAHOO ($\gamma = 10^{-3}$)

(c) COAT ($\alpha = 1$)  (d) YAHOO ($\alpha = 1$)

Figure 3: Test results of MF-CVIB with varying $\alpha$ and $\gamma$. Shaded regions show the $90\%$ confidence intervals of the test AUC.

## 4.4 Results

The evaluation results are reported in Table 2. There are three main observations:

(1) Even if they utilize additional RCTs, the IPS, SNIPS, DR and DRJL methods sometimes work worse than the naive model. By contrast, we identify that our RCT-free CVIB method is capable of enhancing both the shallow and deep models significantly in all experiments. It indicates that the contrastive regularization on the task-aware information, contained in embeddings of factual and counterfactual events, results in improvement of model generalization ability. To further evaluate our method in terms of ranking quality, we report the results of nDCG in Table 3. CVIB shows more significant gain over the baselines than on AUC.

(2) We perform repeated experiments to quantify our CVIB's sensitivity to the balancing term weight $\alpha$ and confident penalty term weight $\gamma$ in Eq. (18). From Fig. 3 we identify that $\alpha$ influences results significantly on COAT, while dose not make much difference on YAHOO. In general, increasing $\alpha$ enhances the test AUC, which is aligned with our argument that contrastive information term benefits in balancing model between factual and counterfactual domains and then leads to better generalization ability. The confidence penalty $\gamma$ plays less role in accuracy, but we should not set it too high to avoid underfitting.

(3) One would notice that the performance of NCF-CVIB in terms of MSE is relatively weak, while we argue that good recommendation does not necessarily rely on low MSE prediction. For instance, for the true outcomes $y = \{1, 0, 0, 0\}$, the predictions $\hat{y}_1 = \{0, 0.1, 0.1, 0.1\}$ have much lower MSE than $\hat{y}_2 = \{0.6, 0.4, 0.4, 0.4\}$, although the former is obviously a worse prediction than the latter in

terms of ranking. It turns out that NCF-CVIB overestimates the outcomes on the test set of YAHOO, nevertheless it still reaches the best ranking quality measured by AUC. In practice of recommender systems, instead of low MSE, we would rather appreciate high ranking quality, namely AUC, which demonstrates the model's capability of finding out the positive examples.

# 5 Related Work

**Counterfactual Learning from MNAR data**. Propensity score based methods have been widely used in causal inference [16], in order to study the treatment effect between different policies. The nature of recommender systems decides that there is large gap between the factual and counterfactual events, i.e., MNAR data by non-uniform policy. In order to remove this sampling bias, inverse propensity score (IPS) method was adopted [28], then followed by doubly robust (DR) [11, 18], weighted DR [34], joint learning DR [37] and self-normalized IPS (SNIPS) [33], while all of these methods require additional RCTs for calculating propensity scores. Besides, recent works proposed to balance factual and counterfactual domains based on RCTs with domain adaptation [19], transfer learning [29], or multi-task paradigm [5]. On contrast, our method can alleviate policy bias by leveraging an information theoretic constraint on representation's information on factual and counterfactual outcomes, which gets rid of the dependence on RCTs.

**Information Theoretic Representation Learning**. Information bottleneck (IB) is conceptually classical in information theory [36], and it has been used in explaining and guiding deep learning practice [3, 31]. The IB garners interests in its application for learning disentangled [2, 14], informative [7, 15], or compressed representation [10]. Above works are extension of the original IB objective loss that is suitable for a specific task. To the best of our knowledge, there is little work on adopting IB in recommendation, especially handling MNAR data. The proposed CVIB loss is based on contrastive information term adapted from the IB loss, which endows its capability of balancing learning with unknown counterfactual outcomes.

# 6 Conclusion

In this work, we proposed an information theoretic counterfactual learning framework in recommender systems. By separating the task-aware sufficiency term in the original information bottleneck objective into factual and counterfactual parts, we derived a contrastive information regularizer and a output confidence penalty, thus obtaining a novel CVIB objective function. The paradigm of information contraction encourages the learned model to be balanced between the factual and counterfactual domains, therefore improves its generalization ability significantly, which is validated by our empirical experiments. Our CVIB provides insight in utilizing information theoretic representation learning methods in recommendation, and sheds light on debiasing learning under MNAR data without performing expensive RCTs.

## Acknowledgements

The research of Shao-Lun Huang was supported in part by the Natural Science Foundation of China under Grant 61807021, in part by the Shenzhen Science and Technology Program under Grant KQTD20170810150821146, and in part by the Innovation and Entrepreneurship Project for Overseas High-Level Talents of Shenzhen under Grant KQJSCX20180327144037831.

## Broader Impact

In this work, we proposed a novel method for dealing with missing-not-at-random feedback in recommendation system. It has been criticized that current recommender systems are prone to overestimating the observed outcomes while underestimating those unobserved yet. As a result, users are restricted to a narrow scope of recommendation results. We would like to propose our method by considering unobserved events, namely counterfactuals, for counterfactual learning on missing-not-at-random feedback. We believe this method brings potential to develop a better recommender system on accuracy as well as diversity and fairness.

## Footnotes

[1]Code is available at https://github.com/RyanWangZf/CVIB-Rec.

[2]Here we slightly abuse notation by denoting $H_{p,q}(y|z) := \mathbb{E}_{x \sim p(x)}\mathbb{E}_{z \sim p(z|x)}\int -p(y|x)\log q(y|z)dy$.

[3]We may also pick $p(y|z^+)$ as the approximation, which renders $H_{p,q}(y|z^+, y|z^-) - H_{p,q}(y|z^+)$. However, it has less operational meaning and its last term cancels out with another task-aware term $I(z^+;y)$.

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
