[Supplementary Material · appendix.pdf]

# A  Proof of Proposition 1

**Proposition 1** (Minimal Representation Insensitive to Policy Bias). *With the Markov chain assumption defined by Eq. (3), for any hidden embedding $h_k$, we can derive the upper bound of the $I(h_k; O)$*

$$I(h_k; O) \leq I(h_k; x) - I(x; y) \leq I(z; x) - I(x; y) \leq I(z; x), \tag{A.1}$$

*where the last term $I(x; y)$ is a constant with respect to the training process.*

*Proof.* From the Data Processing Inequality (DPI) [18], in this Markov chain, we can obtain

$$I(z; x) \geq I(z; y, O) = I(z; O) + I(z; y|O). \tag{A.2}$$

For the second term $I(z; y|O)$, suppose $y$ and $O$ are independent, we can further factorize it and derive

$$I(z; y|O) = H(y|O) - H(y|z, O) \tag{A.3}$$
$$= H(y) - H(y|z, O) \tag{A.4}$$
$$\geq H(y) - H(y|z) \tag{A.5}$$
$$= I(z; y). \tag{A.6}$$

As we assume that $z$ is sufficient, we have $I(z; y) = I(x; y)$. Plugging above result back into Eq.(A.2) yields

$$I(z; x) \geq I(z; O) + I(z; y|O) \tag{A.7}$$
$$\geq I(z; O) + I(z; y) \tag{A.8}$$
$$= I(z; O) + I(x; y), \tag{A.9}$$

which indicates that $I(z; x) - I(x; y)$ bounds $I(z; O)$. And for any hidden embeddings $h_l$, according to DPI, we have

$$I(h_k; z) \leq I(z; x) \quad \forall k \in \{1, \ldots, L\}, \tag{A.10}$$

which yields the final result. $\qquad\square$