[Reviews · NeurIPS 2020]

Review 1

Summary and Contributions: This paper proposes a counterfactual learning framework for use in recommendation systems, where data is typically MNAR (missing not at random). This system is evaluated with two data sets.

Strengths: Recommendation systems typically assume data is MAR; this is a much-needed attempt to explicitly include notions of MNAR data into the literature. The approach is novel and is relevant to the community.

Weaknesses: I would have liked to see evaluation using a ranking-based metric such as NDCG or precision (see below). These are standard in the recommendation systems literature. While the problem is a little different than what is typically seen in recommendation system research (a good thing!), without evaluation with a standard metric, I would be very hesitant to use the proposed method. [1] https://practicalrecs.com/top-model.html [2] Steck, Harald. "Evaluation of recommendations: rating-prediction and ranking." Proceedings of the 7th ACM conference on Recommender systems. 2013. [3] Amatriain, Xavier, et al. "Workshop on Recommendation Utility Evaluation: Beyond RMSE (RUE 2011)." (2012). [4] Cremonesi, Paolo, Yehuda Koren, and Roberto Turrin. "Performance of recommender algorithms on top-n recommendation tasks." Proceedings of the fourth ACM conference on Recommender systems. 2010. [5] Loiacono, Daniele, Andreas Lommatzsch, and Roberto Turrin. "An analysis of the 2014 recsys challenge." Proceedings of the 2014 Recommender Systems Challenge. 2014. 1-6.

Correctness: As far as I can tell.

Clarity: The writing on this paper could be greatly improved. The transitions between idea are sometimes awkward (e.g., starting the second sentence in the abstract with "instead" didn't make sense to me) and some of the points themselves are obtuse on a first read (e.g. line 18/19: "the observed users' feedback can be substantially higher than those not observed yet").

Relation to Prior Work: Dawen Liang has done some work in this space, but nothing by him was cited. For example "Modeling user exposure in recommendation" (https://arxiv.org/pdf/1510.07025.pdf) and maybe "Causal Inference for Recommendation" (http://dawenl.github.io/publications/LiangCB16-causalrec.pdf) come to mind as relevant.

Reproducibility: No

Additional Feedback: Small notes: - line 50: Considering above -> Considering the above - I know it is common to do, but I really don't like using bracketed citations as nouns. E.g., "[9] propose ..." feels wrong. I prefer "Smith et al. [9] propose ..." - line 53: learning representation -> learning a representation - Figure 3 fonts are way too small Research like this is very important, but IMO it just isn't ready yet. I'd rather see it published in a more polished form (writing, evaluation) so it will have more impact. ---UPDATE AFTER AUTHOR FEEDBACK--- Thank you for your detailed responses to the reviews, including new results. In light of this, I am raising my score.


Review 2

Summary and Contributions: In this paper, the authors focus on counterfactual learning in missing-not-at-random data, which is a common and important use case across a variety of domains - including healthcare applications ; though the focus of the paper seems to be on recommendation systems in particular. The authors propose a novel method: information theoretic counterfactual variational information bottleneck (CVIB), enabling the analysis in absence of RCTs. The method aims to learn representations that are equally informative to the factual and counterfactual outcomes. The proposed method is the extension of the information bottleneck (IB) approach. The authors compare the proposed approach to multiple baselines, across two different datasets.

Strengths: The authors propose a novel method for counterfactual learning in absence of RCTs, and the presented results suggest it as a promising.

Weaknesses: The evaluation could be improved in terms of statistical testing in model comparisons, as well as a deeper analysis of failure cases and comparisons between metrics (MSE vs AUC) - as it affects how the results are being interpreted.

Correctness: The claims seem to be correct, given the experiments shown - though additional statistical analysis is needed to support the conclusion of the method significantly improving upon the other baselines - or, alternatively, a more modest phrasing of the conclusions

Clarity: The paper is well written and well structured.

Relation to Prior Work: The authors discuss the relevant related work.

Reproducibility: Yes

Additional Feedback: In this paper, the authors focus on counterfactual learning in missing-not-at-random data, which is a common and important use case across a variety of domains - including healthcare applications ; though the focus of the paper seems to be on recommendation systems in particular. The authors propose a novel method: information theoretic counterfactual variational information bottleneck (CVIB), enabling the analysis in absence of RCTs. The method aims to learn representations that are equally informative to the factual and counterfactual outcomes. The proposed method is the extension of the information bottleneck (IB) approach. The authors compare the proposed approach to multiple baselines, across two different datasets. - Figure 3 a) - it seems like the AUC is increasing with alpha, all the way up to 0.2 - one has to wonder whether it could go even higher beyond 0.2 - did the authors try higher alphas? Would it be possible to extend these plots to show which alpha leads to a maximum AUC? - on the Yahoo dataset (Figure 3 b) ) this is apparent, so the range of values on the x-axis for Figure 3 a) should be extended to the right - otherwise it seems as if the method could conceivably achieve an even higher performance, visually - It seems that the embedding size is not among the hyperparameters being varied in the experiments. Is there a particular reason that it was set to a fixed value of 4? It is mentioned in passing, but it would be good to justify it further in that paragraph, if the choice has been based on a particular observation. Ideally, this point should be extended - as the proposed method is ultimately about representation learning - and may nor work equally well for different embedding sizes. - The authors explain why MSE is not the ideal comparison metric - and that AUC ought to be more important ; to me it felt like this is another place where the paper could be strengthened by doing / showing a bit more of failure case analysis, to quantify different types of errors that the method makes. Especially since CVIB trails heavily in MSE on YAHOO NCF. - COAT data is fairly small (compared to YAHOO) - it’s not entirely clear from the results whether CVIB gives a statistically significant improvement over all other baselines, given that the authors did not compute and include confidence intervals in the table of results. Looking at NCF, COAT AUC - it’s 0.7713 vs 0.7708 of IPS - which does not look very significant. It looks more like a tie - it would be good if the authors could perform statistical testing to only show statistically significant differences. - Having the previous in mind, maybe the following sentence needs a bit more nuance: “By contrast, we identify that our RCT-free CVIB method is capable of enhancing both the shallow and deep models significantly in all experiments” ; significantly has a particular meaning, and without statistical testing should not be used as such ; also, given that the method does not consistently improve MSE, the conclusions are clearly metric-dependent.


Review 3

Summary and Contributions: The paper addresses the missing not at random (MNAR) problem in Recommender System (RS) from a novel perspective – Information Bottleneck (IB). The authors separate user-item pairs (called events) and embeddings into factual and counterfactual parts according to whether the events have been observed. Then they derive a contrastive regularizer, which has a balancing term and a penalty term. The balancing term is used to make the factual and counterfactual embedding be equally informative to the target. The term is counter-intuitive (IMO), but it’s powerful shown in experiments. Whatever, to my best knowledge, this is the first work to solve the MNAR problem in RS by IB.

Strengths: (1) Solving the MNAR problem in RS from a novel perspective. (2) Most of the theoretical derivations are clear and reasonable under their assumptions.

Weaknesses: 1. Some assumptions is not intuitionistic and lack explanation of rationality. Such as in equation (1), how to understand the term “y to x”, where y is true interest and x is event (u,i) instead of interaction? Does y contain any information of x ? Why can take p(y│x^+ ) to approximate p(y|x^+,x^-)? 2. The technical clarity is weak. The main learning framework Equantion (18) is not explained. How to implement the balancing term and the penalty term? 3. IMO, the logic of equation of (13) is not natural. The LHS is always great than or equal to 0, while the RHS is always less than or equal 0. Finding the upper bound of RHS Looks meaningless for the LHS. However, note that the goal is to find a p(z|y) (or say p(z|x), z) to maximize the target term I(y;z)=H(y) – H(y|z), the H(y) should be a constant. So maximizing I(y;z) is equal to maximizing –H(y|z), and then taking the -H_(p,q) (y|z) as a proxy for -H(y|z) would be more reasonable. Meanwhile, the following deriving don’t need to be changed. 4. Lacking some related works, besides IPS- and doubly robust(DR)- based methods, there are some works try to model the mechanism of exposure to solve the same problem, such as [1], and some works try to utilize random-exposure data to solve the problem, such as [2]. [1]. Dawen Liang et.al. Modeling User Exposure in Recommendation. In www, 2016. [2]. Stephen Bonner et.al. Causal Embeddings for Recommendation. In RecSys, 2018. 5. I run the codes on the the Yahoo data to reproduce Table 2. I find the baselines MF is not fairly tuned --- by setting \alpha and \gamma to 0, the method degrades to MF but can reach an AUC of 0.7, which is much higher than the reported score (0.676). The convincingness of experiment results are questionable.

Correctness: There is no problem of the most of theoretical derivation if the assumptions are reasonable. But I am confused about these assumptions. 1. Assumption in equation (1), the relation that “y to x”, where y is true feedback, and y is event (u,i). And the authors assume that the exposure policy is independent with y. Why user-item pairs can be seen as event? If x only represents the user-item pairs, it seems that y don’t contain any information of x. 2. How to understand the z^+ and z^-? Should we take them as a variable in different states (factual and counterfactual), or take them as two variables? Reading the code, I find there is only one embedding layer. So z^+ and z^- should be a same variable but in different states. And in the code, we find x^+ and x^- are randomly selected, I don’t find any relation between the two y in H_(q,q) (q(y│x^+ ),q(y|x^-)). The author use softmax function to process a batch {q(y│x^+ ) } (denote as Q^+=(q_1^+,…,q_n^+)) on the batch dimension, it is hard to understand why do softmax between instances in the batch, but don’t do same operation for q(y|x^-) (similarly denote as Q^-). i.e, Balancing term = 1/N (Q^- )^⊤ log⁡(- softmax(Q^+)), N: batch size Minimizing this term is equal to make the elements of Q^- be small. So the insight of the balancing term is not enough. With this understanding, taking p(y│x^+ ) to approximate p(y|x^+,x^-) is also confusing.

Clarity: - Not good enough. Lack clarity regarding implementation of the proposed methods, explanation of the proposed terms, and analysis of the reasonable of their assumptions.

Relation to Prior Work: This work is a novel method, and different from previous methods.

Reproducibility: Yes

Additional Feedback: Yes, the paper lacking techbnical details, but the authors provide their code. Some questions have been listed in above. Besides above question, I have some additional questions: - Regarding Contrastive Information Regularizer, we can get: I(y;z^+ )-I(y;z^(-1) )=H(y)-H(y│z^+ )-H(y)+H(y│z^- )=H(y│z^- )-H(y|z^+) This form is simpler than the final form proposed in paper. Why not directly take this form? And, we can explain it easily. Such as, minimizing “-H(y|z^+)” is used to constraint the degree of fitting observation data, maximizing H(y│z^- ) is used to make the counterfactual events also be informative (equal to low extropy) - In experiments, embedding size of all methods is set to 4. It’s a small size compared with general setting. Why do you take such small size?


Review 4

Summary and Contributions: The authors study an interesting problem of debiasing the logged feedback via a general information framework. Specifically, the authors propose a framework called counterfactual variational information bottleneck (CVIB) inspired by [16]. Empirical studies on two public datasets show the effectiveness of the proposed framework (using MF and NCF as two backbone methods). Overall, the paper is well presented. For the strengths and weaknesses, please see the comments below.

Strengths: 1 The studied problem of debiasing users’ feedback data in recommender systems is interesting, which is also very important for practitioners in industry. 2 The proposed framework CVIB without using missing at random (MAR) data is novel.

Weaknesses: 1 It seems that the method may not be robust enough. Specifically, when different random seeds are used, different unobserved x^- will be approximated to different observed x^+, which may cause large variance. 2 More explanation and discussions about the fourth term in Equation 18 in Section ion 3.4 shall be included. And I still believe that empirical studies on the fourth term will be helpful. 3 For the rating prediction task (evaluated by MSE), it is not clear why not using the original rating data? 4 For the parameter configurations, it is not explained why set the number of dimensions as 4 (instead of a typical value of 20 or some other values or via grid search)? 5 The baseline methods are not very strong for recommendation tasks of rating prediction or item ranking. There are many matrix factorization based methods or DL methods for these tasks. Thanks for providing the response to some of the raised comments. I think this work is novel and interesting. However, I still believe that the current work is not mature enough for publication in terms of the technical part the empirical results.

Correctness: Correct though I do not check the details very carefully.

Clarity: Yes. Overall, the paper is well presented though there are some grammar errors.

Relation to Prior Work: Right. Technically, it is very clear that the proposed work is different from the existing works.

Reproducibility: Yes

Additional Feedback: N/A

[Author Response · NeurIPS 2020]

**Response for "Information Theoretic Counterfactual Learning from MNAR Feedback"**

We thank the reviewers for their time and for their valuable advice to our work. Due to the space restriction, we will try our best to address their major concerns but we assure that minor comments are also addressed. We would like to stress that this work proposed an information-theoretic method for dealing with MNAR feedback motivated by balancing information of representations. To the best of our knowledge, this is the first time information bottleneck (IB) is adopted in recommendation, and is adapted for **MAR-free** counterfactual learning. We provide an easy-to-implement solution to intractable IB, and verify its effectiveness empirically. It is promising since we prove that without MAR data, using CVIB can reach comparable even superior results to MAR-based methods, since MAR data is really expensive or even impossible to collect in practice.

**To Reviewer #1. (1).** We thank the reviewer for advice to perform evaluation using other metrics used in the literature. We agree with the reviewer and we have evaluated the methods with nDCG using 10 runs: our CVIB shows **more**

| COAT | MF | IPS | SNIPS | DR | DRJL | CVIB | YAHOO | MF | IPS | SNIPS | DR | DRJL | CVIB |
|---|---|---|---|---|---|---|---|---|---|---|---|---|---|
| nDCG@5 | 0.589 | 0.633 | 0.603 | 0.622 | 0.608 | **0.663** | nDCG@5 | 0.633 | 0.636 | 0.635 | 0.659 | 0.652 | **0.734** |
| nDCG@10 | 0.667 | 0.689 | 0.676 | 0.693 | 0.679 | **0.721** | nDCG@10 | 0.762 | 0.760 | 0.762 | 0.774 | 0.770 | **0.820** |

**significant gain** over the baselines than on AUC. We will include these results in the final version of this paper. **(2).** Liang et al. (2015) considered recommendation from positive feedback alone (implicit data); however, we consider learning from both explicit data and unobserved events; Liang et al. (2016) adopted IPS, requiring MAR data as well.

**To Reviewer #2. (1).** When embedding size is 16, MF, IPS, SNIPS, DR and CVIB get AUC 0.67, 0.663, 0.679, 0.667, **0.703** on COAT; and 0.661, 0.648, 0.674, 0.650, **0.717** on YAHOO. Further deterioration appears when embedding size gets larger. It means increasing embedding size **impairs** counterfactual learning due to overfitting, while CVIB still maintains its superiority. **(2).** About MSE. We postulate that the CVIB's weakness in MSE lies in the absence of MAR data in learning. Apart from CVIB, all other baselines utilize MAR data to *regulate the average prediction* to approaching the mean MAR outcomes, by reweighting the loss function. **(3).** About result significance & robustness. We believe that randomness in experiments does exist. For resolving this concern, we try 10 runs of MF, IPS, SNIPS, DR with same learning rate and batch size as MF-CVIB, which yield mean AUC 0.687, 0.704, 0.705, 0.706 on COAT, and 0.678, 0.681, 0.682, 0.684 on YAHOO. On the other, Fig.3 in paper shows the 10 runs of MF-CVIB where it indeed yields better result **on average**: 0.738 on COAT and 0.716 on YAHOO. Interestingly on another ranking metric, e.g., nDCG, CVIB demonstrates more significant advantage. Please refer to the response to **Reviewer #1**.

**To Reviewer #3. (1).** $y \leftrightarrow x \leftrightarrow z$ is a Markov chain in IB by Tishby in 1999 ($\rightarrow$ and $\leftrightarrow$ are just two equivalent marks here). It turns out to obtain compressed $z$ that is predictive to $y$. It is reasonable to assume $x$ contains information of $y$, otherwise by no means we can predict outcome of any event. As mutual information $I(x; y) = I(y; x)$ by definition, $y$ should contain same quantity of information of $x$. **(2).** Since MAR data is absent, there is no way to access the true $p(y|z)$ but only $p(y|z^+)$. This yields $H_{p,q}(y|z^+)$ as the first term in Eq.(15). In fact, we mention the problem in footnote #3 on p.5. Because we use $H_{p,q}(y|z^+)$ as proxy of $I(z^+; y)$ for the sufficiency term, these two terms of $H_{p,q}(y|z^+)$ will *cancel out* in the final objective function (only second term in Eq.(15) plus minimality term in Eq.(18) left). Same result appears if we let $I(y; z^+) - I(y; z^-) = H(y|z^-) - H(y|z^+)$. Overall, using $q(y|z^+)$ instead of $p(y|z^+)$ as approximation is a compromise for ensuring tractability without the loss of empirical performance. **(3).** In Eq.(18), the balancing term is cross entropy between model output $q(y|z^+)$ and $q(y|z^-)$, and the penalty term is entropy of $q(y|z^+)$. They are optimized plus the cross entropy loss between $p(y|z^+)$ and $q(y|z^+)$ together by SGD. **(4).** We factorize $I(z; y) = H(y) - H(y|z)$ hence maximizing $I(z; y)$ is equivalent to minimizing $H(y|z)$, then uses $H_{p,q}(y|z)$ as a proxy. We agree that it would be better to rephrase it as suggested. **(5).** We believe the involved baselines are comprehensive enough, e.g., the strongest baseline in DRJL [26] (ICML2019) is IPS. Stephen Bonner's work is mentioned by reference [10], and please refer to **(2)** to **Reviewer #1** about Liang's works. **(6).** Please refer to **(3)** to **Reviewer #2** about repeat experiments. **(7).** In this work, $z^-$ is embedding of counterfactual event, i.e., $x^-$. And $z^+$ is factual embedding. Given the logged feedback, one event could only belong to either factual or counterfactual set, never both. In this view, minimizing $H_{q,q}(y|z^+, y|z^-)$ amounts to balancing information between factual and counterfactual embeddings on average. Therefore, what we need to do is to sample separately from factual and counterfactual event sets, then optimize on it on average. **(8).** We are sorry that using log softmax is a typo and exaggerates $H_{q,q}(y|z^+, y|z^-)$ thus $\alpha$ in the original experiments should be small. Nonetheless, fixing it then leveling up $\alpha$'s value can result in same performance. To verify this claim, we perform 10 runs on MF-CVIB with $\alpha = 1.0$, same batch size and learning rate. On COAT and YAHOO, the mean and std of AUC is 0.733 (0.007) and 0.719 (0.001), which are even better than the results shown in Table 2. **(9).** Please refer to **(1)** to **Reviewer #2** about the embedding size.

**To Reviewer #4. (1).** We argue that the proposed CVIB is proved robust in this paper, please refer to Fig.3 and **(3)** to **Reviewer #2**. We will show mean/std of other baselines in the final version. **(2).** The fourth term in Eq.(18) is simply $\ell_2$-norm penalty on embeddings, so we add weight decay in ADAM optimizer and find the optimal via grid search. **(3).** Binarizing ratings is commonly used for recommendation. Please refer to section 4.1 in *Causal Inference for Recommendation*. **(4).** Please refer to **(1)** to **Reviewer #2** about the embedding size. **(5).** Although there are many methods in rating prediction, notably few of them are for **counterfactual learning**, i.e., learning from MNAR data and testing on MAR data. Besides, in our experiments, all the selected baselines just use MF & NCF as **backbones**, the same to CVIB for a fair comparison. That means, they are applicable to many backbones, e.g., FM and DeepFM.

[Meta-Review · NeurIPS 2020]

Reviews for this paper were originally quite mixed, and ultimately read as being truly borderline. The positive reviews highlighted some important concerns, but on the other hand the negative reviews didn't raise issues that appeared to be dealbreakers. Some of the main issues identified by reviewers center on a lack of clarity and polish. The positive reviews say the paper is "a good idea, but not ready" (R1), and that the claims need more nuance to be supported (R2). The negative reviews highlight a lack of clarity (R3) and weak baselines (R4), though in the authors' defense these issues are quite fixable. The authors submitted a very detailed response, and a discussion was initiated. After discussing some reviewers changed their reviews, both to increase the scores and to change the tone mostly in a more positive direction, after which I am willing to recommend acceptance.